# Dysregulated Cyclic Nucleotide Metabolism in Alcohol-Associated Steatohepatitis: Implications for Novel Targeted Therapies

**DOI:** 10.3390/biology12101321

**Published:** 2023-10-10

**Authors:** Diego Montoya-Durango, Mary Nancy Walter, Walter Rodriguez, Yali Wang, Julia H. Chariker, Eric C. Rouchka, Claudio Maldonado, Shirish Barve, Craig J. McClain, Leila Gobejishvili

**Affiliations:** 1Department of Physiology, School of Medicine, University of Louisville, Louisville, KY 40290, USA; diego.montoyadurango@louisville.edu (D.M.-D.); mary.walter@louisville.edu (M.N.W.); walter.rodriguez@louisville.edu (W.R.); yali.wang@louisville.edu (Y.W.); claudio.maldonado@louisville.edu (C.M.); 2Department of Neuroscience Training, University of Louisville, Louisville, KY 40290, USA; julia.chariker@louisville.edu; 3KY INBRE Bioinformatics Core, University of Louisville, Louisville, KY 40290, USA; eric.rouchka@louisville.edu; 4Department of Biochemistry and Molecular Genetics, University of Louisville, Louisville, KY 40292, USA; 5Department of Medicine, School of Medicine, University of Louisville, Louisville, KY 40290, USA; shirish.barve@louisville.edu (S.B.); craig.mcclain@louisville.edu (C.J.M.); 6Alcohol Research Center, University of Louisville, Louisville, KY 40290, USA; 7Robley Rex VA Medical Center, Louisville, KY 40206, USA; 8Department of Pharmacology & Toxicology, School of Medicine, University of Louisville, Louisville, KY 40290, USA

**Keywords:** alcohol-associated steatohepatitis, liver, cAMP, cGMP, phosphodiesterases

## Abstract

**Simple Summary:**

Alcohol-associated liver disease (ALD) is a global health problem with high morbidity and mortality. ALD is a multifactorial disease which manifests as lipid accumulation in the liver, hepatic inflammation, fibrosis, and cirrhosis. Due to the complexity of this disease, and despite extensive research to find a cure, there is no FDA-approved therapy to date. Cyclic nucleotides are important second messengers regulating numerous processes in the cells of the body. Their levels and signaling are tightly controlled by sophisticated molecular networks. We have found that cyclic nucleotide levels change in the livers of patients with ALD as well as in mice chronically fed alcohol. These changes are associated with significant alterations in the expression of genes involved in the regulation of cyclic nucleotide signaling and inflammatory/fibrotic processes. Our findings could lead to the development of novel targeted therapies for ALD.

**Abstract:**

Background: Cyclic nucleotides are second messengers, which play significant roles in numerous biological processes. Previous work has shown that cAMP and cGMP signaling regulates various pathways in liver cells, including Kupffer cells, hepatocytes, hepatic stellate cells, and cellular components of hepatic sinusoids. Importantly, it has been shown that cAMP levels and enzymes involved in cAMP homeostasis are affected by alcohol. Although the role of cyclic nucleotide signaling is strongly implicated in several pathological pathways in liver diseases, studies describing the changes in genes regulating cyclic nucleotide metabolism in ALD are lacking. Methods: Male C57B/6 mice were used in an intragastric model of alcohol-associated steatohepatitis (ASH). Liver injury, inflammation, and fibrogenesis were evaluated by measuring plasma levels of injury markers, liver tissue cytokines, and gene expression analyses. Liver transcriptome analysis was performed to examine the effects of alcohol on regulators of cyclic AMP and GMP levels and signaling. cAMP and cGMP levels were measured in mouse livers as well as in livers from healthy human donors and patients with alcohol-associated hepatitis (AH). Results: Our results show significant changes in several phosphodiesterases (PDEs) with specificity to degrade cAMP (Pde4a, Pde4d, and Pde8a) and cGMP (Pde5a, Pde6d, and Pde9a), as well as dual-specificity PDEs (Pde1a and Pde10a) in ASH mouse livers. Adenylyl cyclases (ACs) 7 and 9, which are responsible for cAMP generation, were also affected by alcohol. Importantly, adenosine receptor 1, which has been implicated in the pathogenesis of liver diseases, was significantly increased by alcohol. Adrenoceptors 1 and 3 (Adrb), which couple with stimulatory G protein to regulate cAMP and cGMP signaling, were significantly decreased. Additionally, beta arrestin 2, which interacts with cAMP-specific PDE4D to desensitize G-protein-coupled receptor to generate cAMP, was significantly increased by alcohol. Notably, we observed that cAMP levels are much higher than cGMP levels in the livers of humans and mice; however, alcohol affected them differently. Specifically, cGMP levels were higher in patients with AH and ASH mice livers compared with controls. As expected, these changes in liver cyclic nucleotide signaling were associated with increased inflammation, steatosis, apoptosis, and fibrogenesis. Conclusions: These data strongly implicate dysregulated cAMP and cGMP signaling in the pathogenesis of ASH. Future studies to identify changes in these regulators in a cell-specific manner could lead to the development of novel targeted therapies for ASH.

## 1. Introduction

Alcohol-associated liver disease (ALD) is a global health problem with increasing mortality; this was further exacerbated by the COVID-19 pandemic [1]. It has been reported that of all liver-disease-related deaths, 50% are due to alcohol use [2]. Despite extensive research and clinical trials, there are no FDA-approved therapies to date. ALD is a multifactorial disease characterized by lipid accumulation in the liver (steatosis), inflammation (hepatitis), fibrosis, and cirrhosis [3,4]. Cyclic nucleotides, cyclic adenosine monophosphate (cAMP), and cyclic guanosine monophosphate (cGMP) are important second messengers which play a significant role in signal transduction in every cell in the body. More specifically, cAMP and cGMP signaling modulate numerous processes, including cell proliferation, differentiation, inflammatory response, gut peristalsis, platelet aggregation, and lipolysis, to name a few [5,6]. cAMP is generated in response to external stimuli (hormones, neurotransmitters, and cytokines), which act as ligands to G-protein-coupled receptors (GPCR). Depending on whether this binding activates inhibitory (Gi) or stimulatory G (Gs) proteins will result in either the activation or inhibition of adenylyl cyclases (ACs), enzymes catalyzing the production of cAMP from ATP [7]. The production of cyclic guanosine monophosphate (cGMP), on the other hand, is catalyzed by soluble guanylyl cyclase (sGC) or particulate guanylyl cyclase (pGC) from guanosine triphosphate (GTP) [8]. Four soluble and seven membrane-spanning GCs have been identified in mammals [9]. Atrial natriuretic peptide (ANP), B-type natriuretic peptide (BNP), and nitric oxide are known activators of GCs. The AC family consists of nine transmembrane ACs (AC1-AC9) and one soluble AC (sAC) [10]. An increase in cAMP and cGMP levels leads to allosteric activation of their effectors and downstream signaling. cAMP effector molecules include protein kinase A (PKA), exchange protein directly activated by cAMP (EPAC) [11], Popeye-domain-containing (POPDC) protein [12], and cyclic nucleotide–gated ion channels (CNGCs) [13]. cGMP effectors are represented by the protein kinase G family produced by two genes, PKGI and PCGII [14,15]. 

Phosphodiesterases are a large family of enzymes responsible for cAMP and cGMP hydrolysis to inactive AMP and GMP. Of the eleven members of the family of PDEs, four specifically hydrolyze and degrade cAMP (PDE3, 4, 7, and 9), three degrade cGMP (PDE5, 6, and 9) and others have dual specificity (PDE1, 2, 10, and 11) [16]. Numerous clinical and animal studies have shown beneficial effects of PDE inhibitors in attenuating liver inflammation and fibrosis, strongly suggesting the role of cAMP and cGMP in liver disease pathogenesis (reviewed in [17,18]). Our previous work has shown that dysregulated cAMP signaling plays a critical role in the pathogenesis of alcohol-associated liver disease [19,20,21]. Our first study identified increased PDE4 expression as the underlying alcohol-mediated “priming” of monocytes and macrophages to produce exaggerated levels of inflammatory mediators such as TNF [21]. Later studies have shown that alcohol can affect hepatocytes in a similar fashion and decrease intracellular cAMP levels and signaling [19,20]. Work by others has demonstrated that acetaldehyde-mediated hepatic stellate cell (HSC) activation is regulated by EPAC1 [22]. 

Although the role of cAMP and cGMP signaling is strongly implicated in several pathological pathways in ALD, studies examining the changes in upstream and downstream factors in the metabolism and signaling of these messengers are lacking. Using a mouse model of alcohol-associated steatohepatitis (ASH) developed by Tsukamoto et al. [23], we examined the effect of alcohol on the liver transcriptome to evaluate the changes and regulatory pathways in cAMP and cGMP signaling. We also analyzed publicly available RNAseq data on liver samples of patients with alcohol-associated hepatitis and normal liver tissues from hepatic resection [24]. Our findings could have implications in identifying novel cell-targeted therapies for ALD. 

## 2. Materials and Methods

### 2.1. Human Study

Five liver tissues from healthy donors and six liver tissues (explants) from patients with severe alcohol-associated hepatitis (AH) were obtained from the Resource Center at John Hopkins University (IRB00107893). The mean age of patients with AH was 44 ± 10.1; there were two females and four males. Donors’ gender and age data were not available. All studies were approved by the appropriate Institutional Review Boards and written consent was obtained from all participants.

For the analysis of human samples, raw gene counts for patients with alcohol-associated hepatitis (n = 10) and normal liver tissue from hepatic resection (n = 11) were retrieved from the Gene Expression Omnibus (GSE142530; https://www.ncbi.nlm.nih.gov/geo/query/acc.cgi?acc=GSE142530; accessed on 11 September 2023) and analyzed. 

### 2.2. Animal Experiments

Male C57Bl/6 mice 8 weeks of age were subjected to the alcohol-associated steatohepatitis (ASH) regimen as described [23] (n = 6 pair-fed control and n = 7 alcohol-fed). Briefly, mice were fed a solid Western diet high in cholesterol and saturated fat (HCFD) or regular chow ad libitum for two weeks, followed by the intragastric (iG) feeding of ethanol and a high-fat liquid diet (36%Cal corn oil) at 60% of their total caloric intake plus ad libitum intake of HCFD for the remaining 40% of calories. The ethanol dose was increased to 27 g/kg/day over an eight-week period, and pair-fed (PF) control mice were fed an isocaloric high-fat liquid diet. Additionally, mice were subjected to a weekly alcohol binge (5 g/kg), starting from the second week of iG feeding forward. Mice were euthanized between 10:30 am and 1:00 pm one day after the final binge by inferior vena cava exsanguination, and liver tissues were removed under general anesthesia with Ketamine and Xylazine. All experimental protocols were approved by the University of Southern California Institutional Animal Care and Use Committee (20068), in accordance with the National Institutes of Health Office of Laboratory Animal Welfare Guidelines.

### 2.3. Measurement of Liver Injury Markers 

Plasma alanine aminotransferase (ALT) and aspartate aminotransferase (AST) levels were measured using colorimetric enzymatic assay kits (cat# MAK052 (ALT) and cat# MAK055 (AST), Millipore Sigma, St. Louis, MO, USA), according to the manufacturer’s instructions. 

### 2.4. Cytokine Measurement by MSD Platform

For cytokine measurement, 30–50 mg liver tissue was homogenized in a lysis buffer containing 150 mM NaCl, 20 mM Tris, pH 7.5, 1 mM EDTA, 1 mM EGTA, and 1% Triton X-100 completed with protease and phosphatase inhibitor cocktail. Lysates were incubated overnight on 96-well plates coated with a mixture of antibodies. The plate was read with a MESO QuickPlex SQ 120 imager and analyzed using Discovery Workbench v4.0 software. Data were normalized to the amount of protein per lysate and presented as pg/mg protein.

### 2.5. MPO Activity Assay

After 30–50 mg liver tissue had been homogenized, myeloperoxidase (MPO) activity was measured per the manufacturer’s instructions (HycultBiotech, Wayne, PA, USA). Data were normalized to the protein content in assayed tissue lysate.

### 2.6. Measurements of cAMP and cGMP Levels

cAMP measurements were performed using cAMP complete ELISA (Enzo Life Sciences, Farmingdale, NY, USA, cat# ADI-900-163) and cGMP complete ELISA kits (Enzo Life Sciences, Farmingdale, NY, USA, cat# ADI-900-164). Subsequently, 30–50 mg liver tissue was homogenized in 0.1 N HCl to inhibit phosphodiesterase activity. Homogenates were vortexed and centrifuged for 10 min at >600× *g* to pellet the debris. Levels were normalized by protein content. 

### 2.7. Liver Histopathology and Sirius Red Staining 

Briefly, 5 µm paraffin sections were de-paraffinized in CitriSolv hybrid (Decon Labs, Inc., King of Prussia, PA, USA) and rehydrated in a series of ethanol and water, stained with hematoxylin (Sigma, cat# GHS3) and eosin (Sigma, cat# HT110306) for 1 min each. The sections were then dehydrated and mounted. Sirius Red staining solution was used to stain tissues for 30 min at room temperature. All images were taken using an Olympus BX41 microscope with 10× and 20× objectives.

### 2.8. Statistical Analysis

Statistical analyses were performed using GraphPad Prism version 6.07 for Windows (GraphPad Software Inc., La Jolla, CA, USA). Differences between the two groups were analyzed using the unpaired *t*-test. Results were expressed as the mean ± SD; *p* < 0.05 was considered statistically significant.

### 2.9. RNA Sequencing and Statistical Analysis

Total RNA was isolated from 30–50 mg mouse liver tissues (n = 5 pair-fed and n = 7 alcohol-fed) using TRIzol Reagent (Invitrogen, Carlsbad, CA, USA). The TruSeq Stranded mRNA LT Sample Prep Kit with poly-A enrichment was used to prepare to RNA libraries from 1 μg of sample. Validation of the library was performed qualitatively using an Agilent Bioanalyzer and quantitatively using MiSeq Nano (300 cycles) test runs. Two sequencing runs were performed on the University of Louisville Center for Genetics and Molecular Medicine’s (CGeMM) Illumina NextSeq 500 using the NextSeq 500/550 1 × 75 cycle High Output Kit v2. Quality control was performed using FastQC (v.0.10.1) [25]. Sequenced reads were of good quality, and no sequence trimming was necessary. The sequences were aligned to the mouse reference genome assembly (mm10) using the TopHat2 aligner (v.2.0.13) [26] with alignment rates above 97.5%. Gene read counts were generated using Cuffnorm [27] with FPKM normalization. Differential expression analysis was performed using Cuffdiff2 (v.2.2.1) [27], which utilizes a negative binomial model in the determination of differentially expressed genes. The log2 fold change for differentially expressed genes was input to Pathview [28] for highlighting selected KEGG pathways [29]. 

For the analysis of human samples, raw gene counts for patients with alcohol-associated hepatitis (n = 10) and control patients with a healthy liver (n = 11) were retrieved from the Gene Expression Omnibus (GSE142530) [24]. Differential expression was performed with DESeq2 [30], which uses relative log expression (RLE) as its normalization method. DESeq2 uses a negative binomial model to determine differentially expressed genes, similar to Cuffdiff2 described above.

## 3. Results

### 3.1. Mice Developed Severe Liver Injury, Inflammation, Steatosis, and Pericellular Fibrosis in the ASH Model

Chronic alcohol feeding resulted in significant liver injury and steatosis (Figure 1A). As expected, ASH mice had very high levels of liver enzymes ALT and AST (Table 1). Alcohol-fed mice exhibited significant neutrophil infiltration, resulting in increased MPO activity and increased inflammatory markers (Table 1). There was also evidence of increased stellate cell activation and pericellular collagen deposition typical of ALD fibrosis (Figure 1B). 

RNA sequencing analysis of liver tissues confirmed that inflammatory and fibrogenic pathways were activated in ASH mice (Figure 2A and Figure 2B, respectively). Importantly, comparison of RNAseq findings in mice with those of publicly available AH human liver RNAseq data [24] showed similarities for five proinflammatory genes as well as fifteen genes involved in fibrosis (Figure 2C and Figure 2D, respectively). Interestingly, the expression of interleukin-1 receptor antagonist (*IL1RN*) was decreased in patients with AH, whereas it was increased in ASH mice (Figure 2A,C).

### 3.2. Hepatic Cyclic AMP and Cyclic GMP Levels in ASH Mice and Patients with AH

Our previous studies showed that alcohol decreases cAMP levels in hepatocytes and the whole liver in mice in two different models of ALD [19,20]. Interestingly, we did not observe the same effect of alcohol on liver cAMP levels in this ASH model (Figure 1A). This could be due the differences in models and duration of feeding as well as diet used in this model. In contrast, cGMP levels were significantly increased in ASH mice (Figure 3A). As we have reported previously [20], we did observe much lower cAMP levels in patients with AH compared with healthy donors (Figure 3B). We also found that hepatic cGMP levels were higher in patients with AH (Figure 3B). Notably, baseline levels of cAMP were higher than cGMP in the livers of both mice and humans.

To evaluate how cAMP and cGMP changes and signaling connect to changes in genes and pathways in the liver, we input the log2 fold change for differentially expressed genes to Pathview to highlight selected KEGG pathways (Figure 4). 

### 3.3. Dysregulated Expression of Several Phosphodiesterases by Alcohol in the Livers of ASH Mice and Patients with Alcohol-Associated Hepatitis

To further evaluate the mechanisms of alcohol effects on liver cyclic nucleotide homeostasis, we analyzed the expression levels of PDEs, which are the sole enzymes responsible for their degradation. RNA sequencing detected all PDEs in mouse livers; however, only eight PDEs were affected by alcohol feeding. Among them, we found that three cAMP-specific, two cGMP-specific, and three dual-specificity PDEs were changed in ASH mouse livers (Figure 5a). Notably, among these PDEs, PDE4A, PDE4D, and PDE8A are cAMP-specific; PDE5A and PDE9A are cGMP-specific; and PDE1A, PDE6D, and PDE10A can hydrolyze both cAMP and cGMP (dual-specificity PDEs) [31]. Remarkably, dual-specificity *PDE11A* and *PDE3B*, which are expressed in the human liver at moderate levels [31], were downregulated in AH patients, while all other PDEs were significantly upregulated (Figure 5b). Patients with AH showed a similar alcohol-driven reactivation pattern for five PDEs (*PDE1A*, *PDE4A*, *PDE4D*, *PDE5A*, and *PDE10A*) (Figure 5b), indicating that alcohol-associated disease modulates PDE expression in both murine and human models in a similar fashion. In contrast, other PDEs overexpressed in AH patients but not affected in AF mice included *PDE3A*, *PDE4C*, *PDE6B*, *PDE7A*, and *PDE9A.*

We then examined the enrichment of these genes in various cells in the liver using an open-source online platform, The Human Protein Atlas (THPA, www.proteinatlas.org). We found that these PDEs are expressed in various liver cells, including hepatocytes, hepatic stellate cells, cholangiocytes, T cells, plasma cells, Kupffer cells, etc. (Figure 5c). 

### 3.4. Effect of Chronic Alcohol Feeding on Adenylyl and Guanylyl Cyclases

The effect of alcohol on adenylyl cyclases has been reported before; therefore, we looked at the changes in their expression. We found that alcohol affected the expression of two ACs: AC7 and AC9 (Figure 6a). Notably, AC7 expression was increased while AC9 was downregulated. We also observed the increased expression of two guanylyl cyclases in ASH mice: guanylate cyclase 1 soluble subunit alpha 1 (*Gucy1a1*) and guanylate cyclase 2C (*Gucy2c*). Notably, natriuretic peptide receptors 2 and 3, responsible for cGMP generation, were differentially affected by alcohol feeding (Figure 6a). Analysis of human RNAseq data did not show statistically significant differences for these genes; however, there was increase in AC7 expression in AH patients with the adjusted *p* value of 0.051. GUCY2C expression was also elevated in AH patients, although it did not reach significance (31.3 ± 23.7 in control versus 138.5 ± 86.8 in AH patients, *p* = 0.931). Importantly, these proteins are expressed in various liver cell types (Figure 6b). 

### 3.5. Effect of Alcohol on G-Protein-Coupled Receptor Expression, A-Kinase Anchoring Proteins (AKAPs), and β-Arrestins

The role of GPCR signaling in liver injury and regeneration has been demonstrated in previous studies. Specifically, adenosine receptor 1 (Adora1) has been shown to mediate fat accumulation in the liver [32]. Indeed, we observed that the hepatic expression of Adora1 was increased in ASH mice (Figure 7a). We also observed changes in adrenergic receptors alpha 1b, beta-1, and beta-3 (Figure 7a). Notably, Adora1 receptor signaling is associated with decreased cAMP signaling, while Adra1b and Adrb2 stimulation lead to increased cAMP signaling [33,34]. Importantly, Adrb3 has been shown to produce both cAMP and cGMP [35]. AKAPs play critical roles as scaffolding proteins for the compartmentalized nature of cAMP signaling. Notably, we observed that alcohol feeding increased the expression of *Akap2* and *Akap12* and decreased the levels of *Akap13*. Importantly, alcohol feeding increased β-arrestin 1 and 2 (*Arrb1/2*) expression, which desensitizes GPCRs, suggesting impaired GPCR signaling (Figure 7a). Similarly, livers from patients with AH showed a consistent upregulation of both *ADORA1* and *β-arrestin 2* (*ARRB2*) expression (Figure 7b). However, *ADRB2* was significantly downregulated in AH patients when compared with controls (Figure 7b). Our analysis of human RNAseq data did not show significant changes for the remaining genes analyzed in the mouse model, which could be due to species differences as well as sample size and the severity of disease. Human protein expression databases also show that those proteins, with the exception of Adrb2 and Adrb3, are expressed in both parenchymal and non-parenchymal liver cellular components (Figure 7c). 

### 3.6. ASH/AH Pathogenesis Is Associated with Significant Changes in Cellular Markers Signifying the Changes in Cell Types and Stage of Differentiation

To understand how observed alterations in expression of proteins involved in cAMP/cGMP signaling could impact the cellular processes, we examined the changes in alcohol-induced cellular markers in the liver. We identified more than 30 clusters of differentiation (CD) genes that were significantly affected by chronic alcohol feeding in mice (Figure 8a) and more than 20 CD genes in human livers (Figure 8b). As expected, CD4, which is a marker of T cells, was downregulated in both ASH mice and AH patients, while CD14, a monocyte marker, was upregulated in ASH mice but downregulated in human AH. Markers of B-cells, HSCs, and infiltrated macrophages were also upregulated. Hyaluronan (HA) receptor CD44, which is mainly expressed on HSCs in the liver [36,37] and has been shown to drive fibrosis, was also increased. CD63, CD68, CD9, and CD5l (hepatic macrophage markers) were increased in ASH mice livers, while CD59a/b, CD79a/b, CD177, and CD302 were downregulated. Importantly, CD59 and CD302 have been shown to be expressed not only on immune cells, but also on hepatocytes [38,39,40]. CD79a/b is a marker of B cells [41,42], while CD177 is expressed in neutrophils [43]. 

The fibrosis marker CD44 was also consistently upregulated in AH (Figure 8b), as was the tetraspanin CD63 [44], and CD151 [45], important proteins for the function of B cells and dendritic cells. However, in comparison with ASH livers, CD36, a protein involved in the uptake of fatty acids by hepatocytes and a direct contributor to fatty liver, was downregulated [46] in AH livers. Similarly, CD5L, a circulating protein that protects hepatocytes from excess fat accumulation and malignancy [47], was downregulated in AH livers (Figure 8b). Overall, these data show that chronic alcohol feeding leads to changes in various parenchymal and nonparenchymal cell numbers and their function. Given the importance of cAMP/cGMP signaling in cell function, these data suggest that alcohol-mediated changes in these cyclic nucleotides contributed to the alterations we observed in cellular markers. At the same time, it could also explain the changes observed in the expression of PDEs and AKAPs as other regulatory proteins of cyclic nucleotide signaling.

## 4. Discussion

The roles of cAMP and cGMP as critical second messengers in cell function and response are well established, and the effect of alcohol on dysregulated cAMP signaling is also well documented (reviewed in [17]). Initial studies in peripheral blood mononuclear cells have reported decreased cAMP levels and signaling in patients with alcohol-associated hepatitis [48]. Further study of lymphocytes in patients with ALD showed lower basal and adenosine-induced cAMP levels [49]. This effect is mediated by the desensitization of GPCR coupled with stimulatory G protein, Gs [50]. Moreover, chronic alcohol exposure decreases Gsα expression at both mRNA and protein levels [50]. In hepatocytes, acute alcohol exposure showed a biphasic, dose-dependent effect on cAMP production in response to glucagon without any effect on adenylyl cyclase activity [51]. Our own studies demonstrated that chronic alcohol exposure decreased cAMP levels in monocytes and macrophages, including in Kupffer cells isolated from alcohol-fed rats [52]. We later showed that this effect was driven by the increased expression and activity of PDE4B [21]. In hepatocytes, our studies showed that alcohol increased PDE4A, B and D mRNA, and protein expression in primary rat and mouse hepatocytes and whole livers after alcohol feeding and decreased cAMP levels [19,20]. Interestingly, we did not observe the same decrease in cAMP levels in mouse livers with ASH in this study. Similar results were reported in a study using intragastric alcohol feeding of rats for two months [53]. However, the same study showed that administration of a cAMP analog had a beneficial effect on liver injury [53]. Using the same model of the intragastric feeding of rats, another study showed that liver regeneration was impaired by alcohol feeding due to decreased adenylyl cyclase activation. They also noted a decreased expression of stimulatory G protein, Gs and an increased expression of inhibitory Gi2α after partial hepatectomy [54]. These results agree with our observations that alcohol affected the expression of GPCRs, which activated Gi and decreased Gs (Figure 4A). 

The effect of alcohol on adenylyl cyclase 7 is well studied in patients with alcohol use disorder (AUD) [55]. Specifically, it has been shown that alcohol increases AC7 activity in the brain tissue of patients with AUD [56]. We found a similar effect of alcohol on AC7 (*Adcy7*) mRNA levels in the liver (Figure 6A). However, AC7 is expressed in various liver cell types, including hepatocytes, HSCs, and Kupffer cells (KCs) (Figure 6B). Hence, it is likely that several cell types are affected. Notably, we observed a decrease in *Adcy9* caused by alcohol, which is expressed in the liver in hepatocytes, KCs, plasma, and vascular endothelial cells. Future studies are needed to determine both the target cells as well as the impact at the cellular and functional levels.

Numerous studies have demonstrated that the activation of cyclic nucleotide signaling by inhibiting PDEs is beneficial for various liver injury/fibrosis models (reviewed in [17,18]). Our own studies have shown that upregulation of the cAMP-specific PDE4 family of enzymes is associated with the development of cholestatic liver injury in rats [57]. This upregulation is accompanied by liver inflammation, injury, and fibrogenesis in rats. Notably, we showed that PDE4 plays a role in the spontaneous differentiation of hepatic stellate cells (which are major contributors to liver fibrosis [57]). Our recent work further demonstrated that PDE4D enzymes are expressed in activated HSCs in human and mouse livers and play a role in promoting cytoskeleton remodeling and HSC migration [58]. Notably, PDE1a has also been shown to regulate cell motility and the migration of cancer cells via cGMP/PKG [59]. Relevant to ALD, our studies identified the PDE4B-dependent downregulation of cAMP signaling as a pathogenic player in exaggerated responses of monocytes/macrophages to endotoxin on ALD [21,52]. Subsequent studies have also identified PDE4 as a player in alcohol-induced dysregulated lipid metabolism and injury in the liver [19,20]. A more recent study showed that overexpression of PDE4D in the liver led to the development of NAFLD and hypertension in mice, which was attenuated by PDE4 inhibitor treatment [60]. Beyond PDE4, recent papers have shown a role of PDE9 and 10 in liver and lung fibrosis as well as diet-induced obesity [61,62,63,64]. Our data show a significant upregulation of PDE9A and 10A in the livers of AH patients (Figure 5). Importantly, the exact role of these PDEs in ALD is unknown and needs to be investigated. 

Immune cell infiltration to the liver and persistent inflammation are major drivers of liver injury and fibrosis. In addition to resident Kupffer cells, which produce profibrogenic TGFβ1, infiltrated monocytes, T cells, and neutrophils play roles in the perpetuation of inflammation and fibrosis [65]. cAMP signaling via protein kinase A negatively regulates T cell activation, monocyte adhesion/migration, and neutrophil function [66,67,68]. Conversely, activation of these immune cells is associated with the increased activity of cAMP degrading PDE4. Indeed, PDE4B and PDE4D were found to be overexpressed in PBMCs from patients with Psoriasis [69], an inflammatory disease characterized by the dysfunction of dermal fibroblasts. The deletion of *Pde4b* and *Pde4d* has been shown to decrease neutrophil adhesion molecule expression and chemotaxis, the production of TNF, and T cell activation [70,71,72]. It has also been demonstrated that deletion of *Pde4b* significantly reduces endotoxin (LPS)-inducible TNF production in circulating leukocytes and macrophages [21,73,74,75]. PDE4 inhibitor has also been shown to reduce leukocyte infiltration, oxidative processes, and tissue damage in animal models [20,70,76]. Concerning T cell function, it has been shown that PDE8, a cAMP-specific PDE, plays a critical role in T cell recruitment and function in inflammation [77]. Specifically, PDE8A has been shown to regulate T cell motility and adhesion, unlike PDE4 [77]. Notably, PDE8 expression has been identified in highly purified CD4^+^T cells in vivo, which was significantly increased upon activation (reviewed in [77]). Our results are consistent with these studies. We found that PDE8A was downregulated in ASH mice livers along with significantly decreased levels of CD4, a cell marker of CD4^+^T cells. Notably, CD4^+^T cells are affected by alcohol and contribute to alcohol-mediated immunosuppression [78,79,80]. 

PDE1, 2, 3, 4, and 5 enzymes play critical roles in vascular function via regulating cAMP and cGMP in endothelial [81] and smooth muscle cells [82]. These functions include vascular tone, exchange, and remodeling. Nitric oxide–cGMP signaling is crucial in regulating hepatic sinusoids and portal pressure. Our data demonstrate that alcohol upregulated PDE1A, 4A/D, and 5A in both mouse and human livers. These changes in vascular and smooth muscle cells could lead to significant changes in blood flow and vascular permeability. In fact, as with our findings, it has been reported that sGC and PDE5 are overexpressed in cirrhotic livers [83]. It was also proposed that targeting soluble guanylate cyclases (sGC) and PDE5, and modulation of the cGMP pathway, is beneficial for portal hypertension during cirrhosis. Notably, one study examined the cellular expression of PDE5 protein and found that in the normal liver it is weakly expressed in hepatocytes and highly expressed in perisinusoidal cells. However, in cirrhotic livers, PDE5 expression is evident in fibrous septa, hepatocytes adjacent to veins, and perisinusoidal cells throughout the parenchyma [83]. Animal studies have demonstrated sGC in HSCs and myofibroblast in fibrotic livers and human NASH (reviewed in [83]). sGC stimulation and PDE5 inhibition shown anti-inflammatory and antifibrotic effects by normalizing cGMP levels [84,85,86]. Importantly, we show, for the first time, that cGMP levels are elevated in the livers of patients with AH (Figure 3B). Notably, plasma levels of cGMP are increased in patients with cirrhosis, including in those with ALD cirrhosis [87,88,89]. In fact, increased cGMP levels were proposed to be used as a marker of portal hypertension in patients with liver cirrhosis [83]. It is important to highlight that age-dependent sexual dimorphism in the expression patterns of vascular PDEs has recently been reported [90]. Hence, we expect differences in liver PDEs in males and females and therefore a differential regulation of cellular processes by cAMP/cGMP.

A-kinase anchoring proteins (AKAPs) represent a family of proteins whose function is to serve as scaffolding proteins for the specificity and spatiotemporal nature of cyclic nucleotide signaling [91,92,93]. They achieve this specificity by assembling multiprotein signaling complexes consisting of PKA, PDEs, and phosphatases [93]. We found that alcohol changed the expression of three mouse AKAPs: AKAP2 (increased), AKAP12 (increased), and AKAP13 (decreased). Notably, AKAP13 has been recently described as a regulator of GPCR-mediated inhibition of mTORC1 signaling by acting as a scaffold for PKA and mTORC1 [94]. Hence, the observed decrease in AKAP13 in ASH may contribute to increased mTORC signaling, which plays a causal role in ALD development [95,96]. The role of AKAP12 in liver injury and fibrosis has also been described [97,98,99,100]. Less is known about the role of AKAP2 in the liver, but it seems to be enriched in immune cells. In contrast, our analysis of AH patient samples did not show significant changes for AKAPs, although it exhibited increased expression of *ADORA1* and *ARRB2*, genes involved in alcohol-induced hepatic steatosis [32] and apoptosis [101]. Future studies are needed to evaluate the exact role of these AKAPs in liver cells and ALD pathogenesis. 

## 5. Conclusions

Despite the experimental evidence of dysregulated cAMP/cGMP signaling in the development of liver pathologies, including ALD, a comprehensive examination of the factors contributing to the signaling of these critical messengers has not been undertaken. In this study, we attempted to capture the changes in regulatory gene networks involved in cAMP/cGMP signaling imposed by chronic alcohol consumption. Importantly, the intragastric alcohol feeding model used in this study is the best animal model to recapitulate the spectrum of ALD, specifically steatosis, inflammation, and fibrogenesis. Furthermore, we compared the gene expression profiles with publicly available human transcriptome data for AH patients and identified similar patterns of altered gene expression. Our data strongly implicate dysregulated cAMP and cGMP signaling in the pathogenesis of ASH. Future studies to identify changes in these regulators in a cell-specific manner could lead to the development of novel targeted therapies for ASH. 

## Figures and Tables

**Figure 1 biology-12-01321-f001:**
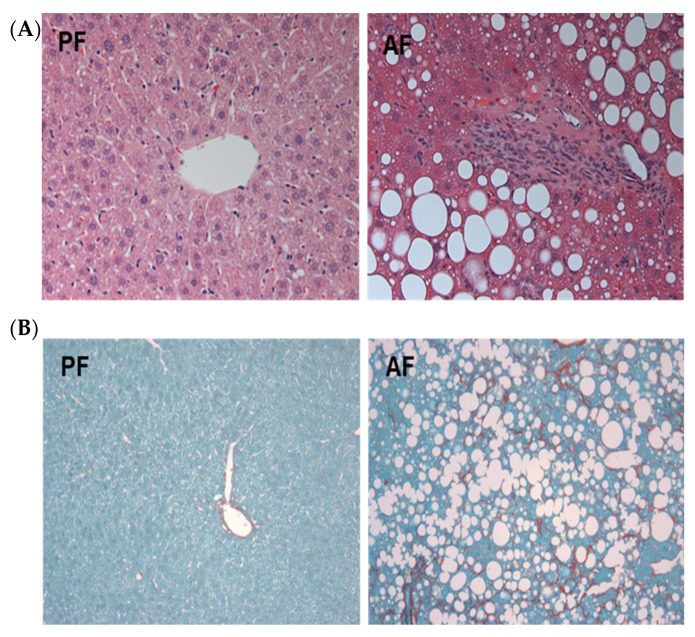
Alcohol-fed mice develop severe hepatic steatosis and pericellular fibrosis. (**A**) Paraffin-embedded liver sections were stained with hematoxylin and eosin. Representative images of liver tissues from pair-fed (PF) and alcohol-fed (AF) mice subjected to intragastric feeding as described in the Section 2. (**B**). Sirius Red staining showing pericellular collagen deposition in the livers of AF mice as compared with PF counterparts, demonstrating fibrotic processes in alcohol-fed mice.

**Figure 2 biology-12-01321-f002:**
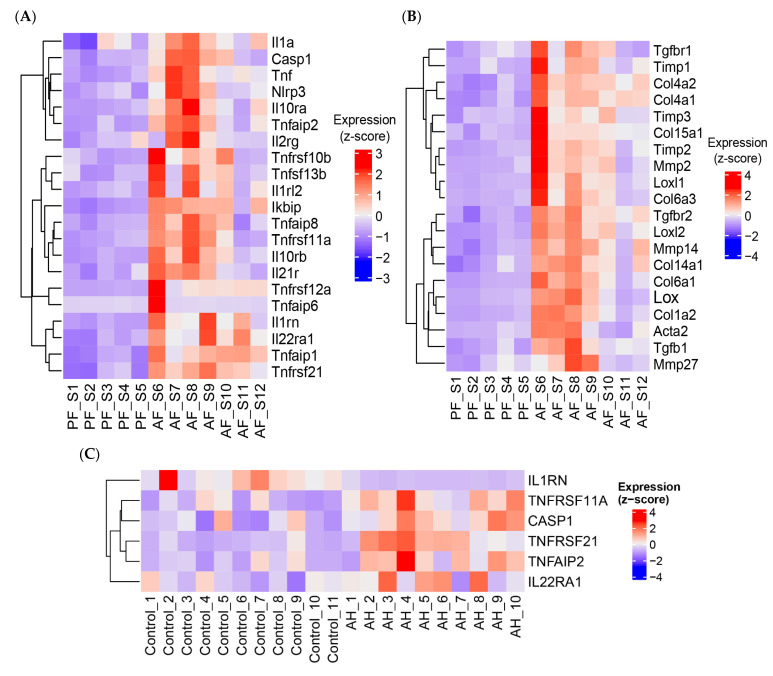
Heatmaps of significantly upregulated (**A**) inflammatory and (**B**) fibrotic genes in mice fed alcohol when compared with their PF counterparts. (**C**,**D**). Reciprocal analysis was performed using liver RNAseq data from human liver samples [24], from patients with alcohol-associated hepatitis (AH), and normal liver tissues from hepatic resection (control). Hierarchical clustering was performed on differentially expressed genes following Cuffdiff2 and log2 analysis (for murine model) or DESeq2 for human samples.

**Figure 3 biology-12-01321-f003:**
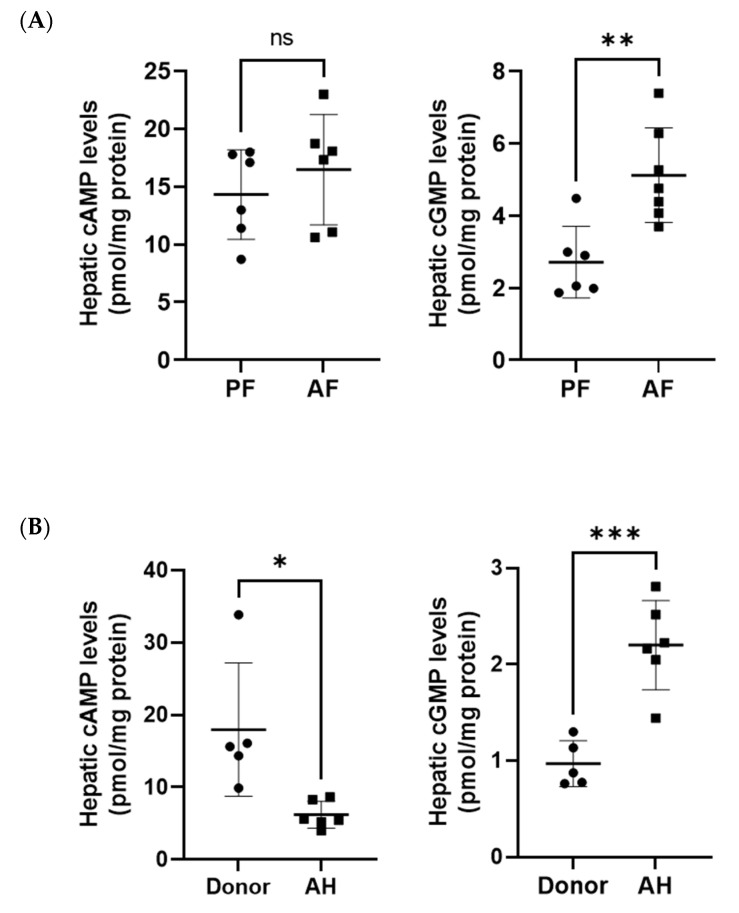
Hepatic levels of cAMP and cGMP. (**A**) cAMP and cGMP levels were measured in liver tissues of PF and AF mice, n = 6–7. (**B**) cAMP and cGMP levels in liver tissues from human healthy donors and patients with alcohol-associated hepatitis (AH), n = 5–6. Data are presented as the mean ± standard deviation, * *p* < 0.05, ** *p* < 0.01, *** *p* < 0.001, ns not significant.

**Figure 4 biology-12-01321-f004:**
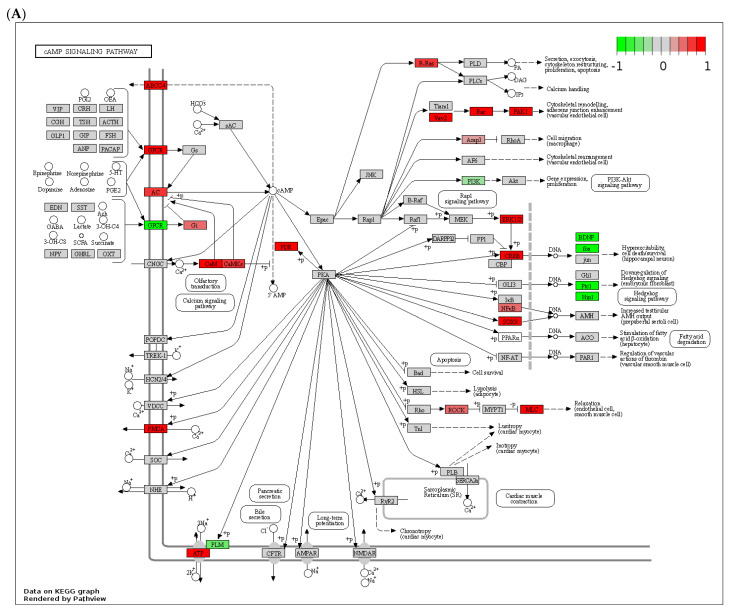
(**A**) cAMP-PKA and (**B**) cGMP-PKG signaling pathways highlighted with significant upregulated (red) and downregulated (green) genes using RNAseq data from the mouse ASH model.

**Figure 5 biology-12-01321-f005:**
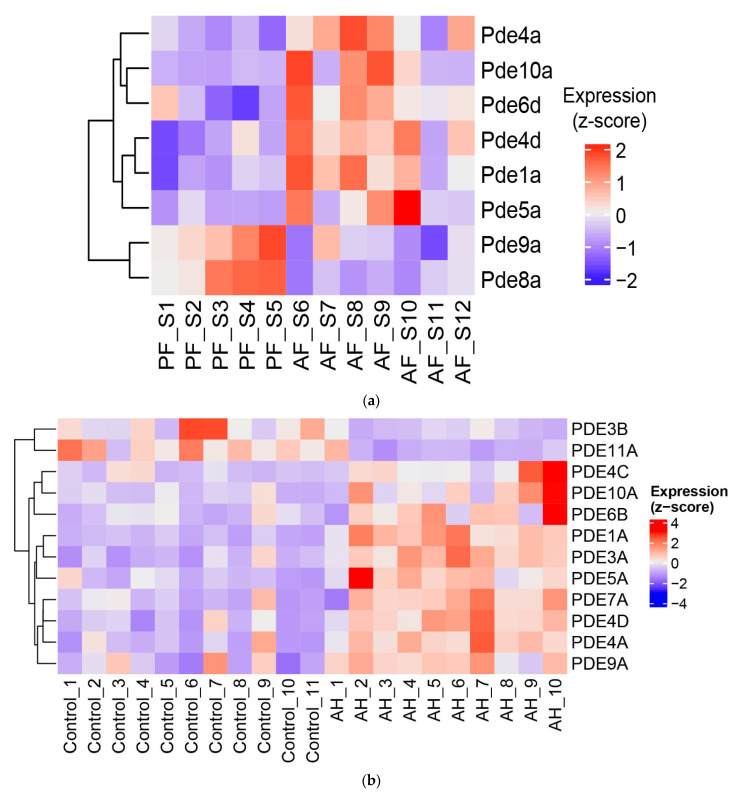
(**a**) Heatmap of PDE genes significantly altered by alcohol feeding in the livers of mice: PF, pair-fed; AF, alcohol-fed. (**b**) Heatmap of significantly altered PDE genes for human liver samples: AH, alcohol-associated hepatitis; control, normal liver tissues from hepatic resection. (**c**) Reported expression patterns of each PDEs in various liver cell types, derived from https://www.proteinatlas.org.

**Figure 6 biology-12-01321-f006:**
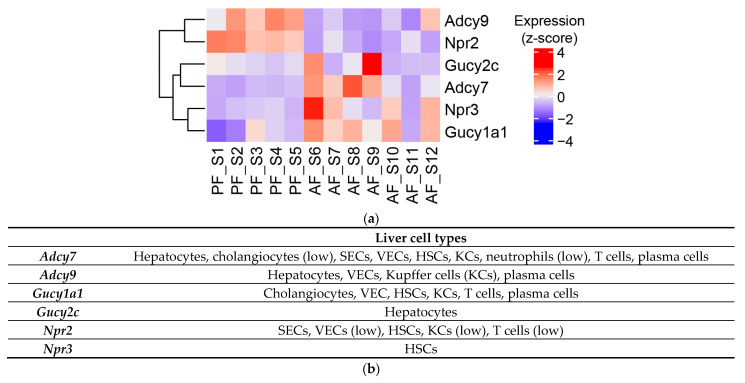
(**a**) Heatmap of adenylyl and guanylyl cyclase genes significantly altered by alcohol feeding in the livers of mice. (**b**) Reported expression patterns in liver cells derived from https://www.proteinatlas.org.

**Figure 7 biology-12-01321-f007:**
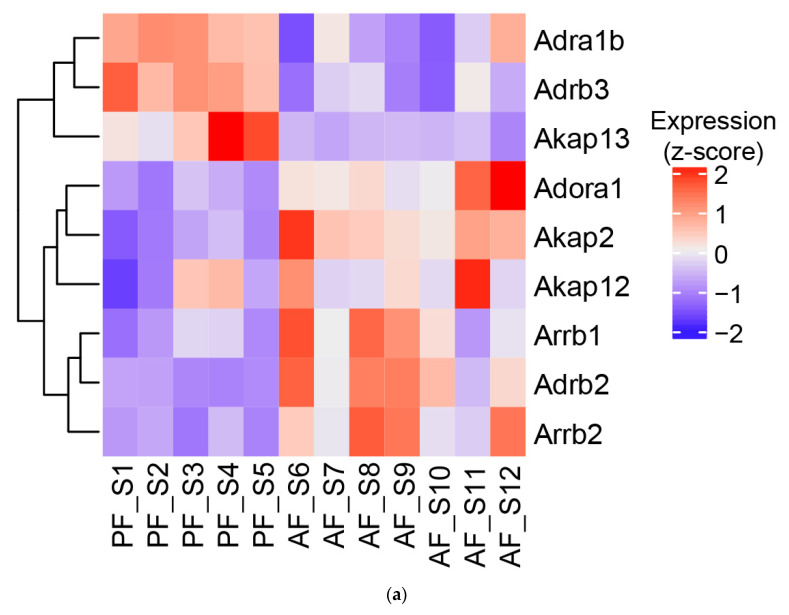
(**a**) Heatmap of genes significantly altered by alcohol feeding in mice. (**b**) Heatmap of RNAseq data for human liver samples from patients with alcohol-associated hepatitis (AH) and control—normal liver tissues from hepatic resection. (**c**) Reported expression patterns of each gene in liver cells derived from https://www.proteinatlas.org.

**Figure 8 biology-12-01321-f008:**
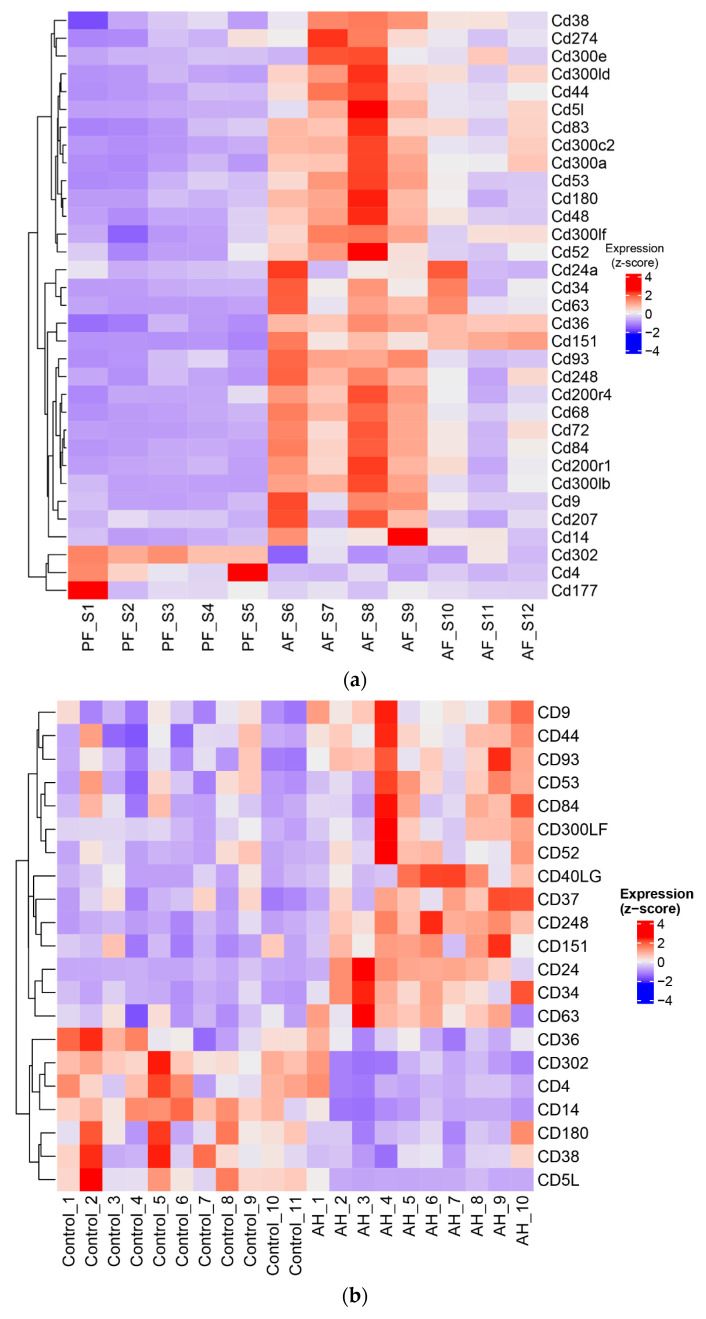
Heatmap showing a significant effect of alcohol on cellular markers in (**a**) mouse livers: PF, pair-fed; AF, alcohol-fed. (**b**) Heatmap showing a significant effect of alcohol on cellular markers in liver samples from patients with alcoholic hepatitis (AH) and control—normal liver tissues from hepatic resection.

**Table 1 biology-12-01321-t001:** Plasma levels of liver injury markers ALT and AST, hepatic MPO activity, and cytokine levels in mice.

	Pair-Fed	Alcohol-Fed
ALT, U/L	15 ± 5.8	348.3 ± 169.01 **
AST, U/L	53.5 ± 8.9	289 ± 190.4 *
MPO activity, Units/mg	9.6 ± 1.97	32.4 ± 16.68 *
TNFα, pg/g	4.3 ± 2.63	10 ± 3.8 *
IP-10, pg/g	156.8 ± 25.69	367.5 ± 117.97 **
KC-GRO, pg/g	155.3 ± 27.18	362.7 ± 106.15 **
GM-CSF, pg/g	0.15 ± 0.05	0.3 ± 0.08 *

Data are presented as the mean ± standard deviation, n = 5 in pair-fed, n = 7 in alcohol-fed groups, * *p* < 0.05, ** *p* < 0.01.

## Data Availability

The data generated or analyzed during this study are included in this published article.

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
