# Peer review of "Dysregulated Cyclic Nucleotide Metabolism in Alcohol-Associated Steatohepatitis: Implications for Novel Targeted Therapies"

_biology, 2023, doi:10.3390/biology12101321_

Round 1

Reviewer 1 Report

The authors examined a potential role of cyclic nucleotides (cAMP / cGMP) metabolizing genes in the pathogenesis of ALD. In general, experiments are well designed and the manuscript demonstrates a high level of structural integrity, effectively providing essential information to the readers in a concise and coherent manner. The animal model used in this study is sound relevant to human ALD development and pathology. However, I have few minor comments to enhance the quality of the manuscript and better serve the audience.

 Authors analyzed both human and mouse tissues in this manuscript. Please specify whether the samples analyzed are human or mouse in the figure legends. For example, in Fig. 5A, 6A, 7A & 8, clarify whether the data pertains to human or mouse samples.

 The expression of several PDEs was found to be affected in AF mice livers. It would be valuable to determine whether these findings are consistent in humans. Therefore, it is suggested that the authors consider assessing the expression of these PDEs in patients with AH (Alcoholic Hepatitis). This additional analysis could provide important insights into the relevance of their findings in a clinical context.

 Figure 7B is not referenced or cited anywhere in the manuscript text.

Author Response

Dear Editor,

We thank reviewers for their constructive comments and suggestions. We have expanded the overall reach and depth of the manuscript by adding new data demonstrating gene expression changes related to cyclic nucleotide signaling in relevant human liver tissues. These data were generated by analyzing publicly available RNAseq databases. In this revised manuscript, we are presenting data from mouse and human studies side-by-side to demonstrate clinical significance of cyclic nucleotide signaling in the pathogenesis of ALD. New figures include human data on gene expression changes in both inflammatory and fibrotic targets (Figure 2), phosphodiesterase family (PDE) gene profiles (Figure 5), adenylyl and guanylyl cyclase genes (Figure 6), GPCR-associated genes (Figure 7), and both parenchymal and non-parenchymal markers of liver cell populations (Figure 8). The results section has been revised accordingly with new figures and respective figure legends, and the discussion reflects the changes in the results section. Likewise, the methods section has been updated to incorporate the changes described above.

All the appropriate changes are included in the revised version of the manuscript (highlighted in yellow).  We hope the revised version of the manuscript, which systematically addresses the comments made by the reviewers, will be acceptable for publication.

Following is the point-by-point response to comments from Reviewers:

Reviewer #1

“Authors analyzed both human and mouse tissues in this manuscript. Please specify whether the samples analyzed are human or mouse in the figure legends. For example, in Fig. 5A, 6A, 7A & 8, clarify whether the data pertains to human or mouse samples.”

Response: We have updated all the figure legends in the manuscript to clarify both the source (mouse or human), as well as the type of sample (plasma, liver homogenate, or RNA from liver tissue).

“The expression of several PDEs was found to be affected in AF mice livers. It would be valuable to determine whether these findings are consistent in humans. Therefore, it is suggested that the authors consider assessing the expression of these PDEs in patients with AH (Alcoholic Hepatitis). This additional analysis could provide important insights into the relevance of their findings in a clinical context.”

Response: We thank the reviewer for this important comment. To address this question we downloaded and analyzed a publicly available liver RNAseq data for human control and AH patients (Gene Expression Omnibus (GSE142530); for reference see Gastroenterology, Volume 160, Issue 5, 2021, Pages 1725-1740.e2, https://doi.org/10.1053/j.gastro.2020.12.008).  The revised manuscript incorporates the analysis and comparison between the murine ASH model and the human AH patients for the target genes analyzed by RNAseq in figures 2, 5, 6, 7, and 8. Of note, a large number of genes showed a strikingly similar gene expression pattern in both human and mouse livers, thereby strengthening the value of the ASH model for understanding the alcohol-associated liver disease process and target genes in humans that could be of importance as druggable alternatives.

“Figure 7B is not referenced or cited anywhere in the manuscript text.”

Response: We have added new panels to Figure 7. Hence, the former Figure 7B is now Figure 7C.  We added the following text: “Human protein expression databases also show that those proteins, with exception of Adrb2 and Adrb3 are expressed in both parenchymal and non-parenchymal liver cellular components (Figure 7C).”

Reviewer 2 Report

In the manuscript entitled “Dysregulated cyclic nucleotide metabolism in alcohol associated steatohepatitis: implications for novel targeted therapies” by Montoya-Durango et al., the authors have studied the role of AMP/cGMP signaling (cyclic nucleotides) which acts as secondary messengers in the development of liver pathologies including ALD. Using male C57B/6 mice they developed an intragastric model of alcohol-associated steatohepatitis (ASH). Liver injury was confirmed by evaluating markers of inflammation and fibrogenesis by measuring plasma levels of injury markers, liver tissue cytokines and gene expression analyses.  Furthermore, liver transcriptome analysis was performed to determine the role of cAMP and cGMP signaling pathways during alcohol-induced liver injury. Liver samples from human healthy donors and patients with alcohol-associated hepatitis (AH) were also included in the study to determine the role of cAMP and cGMP signaling pathways during ALD. To understand the role of cAMP and cGMP pathways, mRNA-based sequencing analysis was performed which revealed dysregulation of several enzymes/ proteins that regulate cAMP and cGMP levels including phosphodiesterases (PDE), adenylyl cyclases,  adenosine receptor 1, beta-arrestin 2. Taken together, the present manuscript demonstrates dysregulation of cAMP and cGMP signaling during the pathogenesis of ASH. Further, authors have demonstrated increased inflammation, steatosis, apoptosis and fibrogenesis in the model of ALD. Overall, the study is interesting and well presented, however, following suggestions will help to improve the manuscript.

1. The authors have used human liver samples (control and patient), it is important to provide the details of inclusion and exclusion criteria in the present study. Also, what were the pathologies reported, co-morbidities present and other clinical information available regarding the patients? What was their history of alcohol consumption? What was the mean age in human controls?

2. Kindly mention whether the protocol for animal studies was approved by the animal ethical committee.

3. How the mice were euthanized? What was the age of mice that were used to make the model? Also, control mice, who received a normal diet shall be included in the study. Further details such as sleep cycle and other conditions for the mice must be mentioned.

6. In the material and method section, the protocols are superficial. Overall, the protocols need more clarity, for example, how much tissue was taken (in mg), protein? How many times the experiments were repeated? The statistics used also need more clarity, as some of the experiments required 2-way ANOVA and other statistical analysis(bioinformatics, mRNA sequencing data).

9. For RNA study, how many tissues were taken? How many mice were used for mRNA sequencing? Validation of these markers (identified through sequencing) is also required, preferably in mice as well as human tissue.

10. In the material and method section, the details about bioinformatics analysis shall be provided.

11. Caption of figures must be self-explanatory. Currently, the caption of figures is very superficial.

Can be improved, otherwise, the manuscript is written clearly.

Author Response

We thank reviewers for their constructive comments and suggestions. We have expanded the overall reach and depth of the manuscript by adding new data demonstrating gene expression changes related to cyclic nucleotide signaling in relevant human liver tissues. These data were generated by analyzing publicly available RNAseq databases. In this revised manuscript, we are presenting data from mouse and human studies side-by-side to demonstrate clinical significance of cyclic nucleotide signaling in the pathogenesis of ALD. New figures include human data on gene expression changes in both inflammatory and fibrotic targets (Figure 2), phosphodiesterase family (PDE) gene profiles (Figure 5), adenylyl and guanylyl cyclase genes (Figure 6), GPCR-associated genes (Figure 7), and both parenchymal and non-parenchymal markers of liver cell populations (Figure 8). The results section has been revised accordingly with new figures and respective figure legends, and the discussion reflects the changes in the results section. Likewise, the methods section has been updated to incorporate the changes described above.

All the appropriate changes are included in the revised version of the manuscript (highlighted in yellow).  We hope the revised version of the manuscript, which systematically addresses the comments made by the reviewers, will be acceptable for publication.

Following is the point-by-point response to comments from Reviewers:

Reviewer #2

  1. The authors have used human liver samples (control and patient), it is important to provide the details of inclusion and exclusion criteria in the present study. Also, what were the pathologies reported, co-morbidities present and other clinical information available regarding the patients? What was their history of alcohol consumption? What was the mean age in human controls?”

Response: Human liver samples used for this study were obtained from Johns Hopkins University as part of an NIH funded grant, “Clinical Resources for Alcoholic Hepatitis Investigations” (project number 5R24AA025017-05). These resources include livers and data from patients with severe AH during transplantation, and wedge biopsies from donor livers as controls. The goal of this project is to “establish a centralized database of de-identified samples for the purpose of promoting access to otherwise unavailable specimens, collaboration, efficiency, and progress towards a cure of AH”. We already submitted RESEARCH PARTICIPANT INFORMED CONSENT AND PRIVACY AUTHORIZATION FORM provided by Dr. Zhaoli Sun, who is principal investigator of this grant. The information on donors is not available, as we mentioned in the manuscript. We already provided age and sex of AH patients.

“2. Kindly mention whether the protocol for animal studies was approved by the animal ethical committee.”

Response: We have incorporated the following sentence in section 2.2, Animal experiments: “All experimental protocols were approved by the University of Southern California Institutional Animal Care and Use Committee in accordance with the National Institutes of Health Office of Laboratory Animal Welfare Guidelines.”

“3. How the mice were euthanized? What was the age of mice that were used to make the model? Also, control mice, who received a normal diet shall be included in the study. Further details such as sleep cycle and other conditions for the mice must be mentioned.”

Response: The mouse ASH model was developed and executed by the Animal Core of the Southern California Research Center for ALPD and Cirrhosis (PI: Dr. Tsukamoto). Age of mice before starting the experiment was 8 weeks. Mice were euthanized between 10:30am to 1pm one day after the final binge by inferior vena cava exsanguination and liver tissues were removed under general anesthesia with Ketamine and Xylazine. We included all this information in Methods section.

Regarding the control mice requested by the Reviewer, all ALD models use pair-fed mice as “controls” to their alcohol fed counterparts, which allows for controlling the total daily caloric intake.

“6. In the material and method section, the protocols are superficial. Overall, the protocols need more clarity, for example, how much tissue was taken (in mg), protein? How many times the experiments were repeated? The statistics used also need more clarity, as some of the experiments required 2-way ANOVA and other statistical analysis (bioinformatics, mRNA sequencing data).”

Response: We have revised the section and added information on statistics as well as amount of tissue used.

“9. For RNA study, how many tissues were taken? How many mice were used for mRNA sequencing? Validation of these markers (identified through sequencing) is also required, preferably in mice as well as human tissue.”

Response: We have added the number of mice used for each experimental group for RNA sequencing, however our heatmaps already show how many mice are in each group (n=5 in pair-fed (PF) and n=7 in alcohol-fed (AF). As for validation of these markers, we have performed real time qPCR analyses for many of the genes we present to validate the findings. Some of hepatic inflammatory markers were measured by MSD milliplex ELISA-type assay, and results are presented in Table 1 (TNF, IP-10, KC-GRO and GM-CSF). Additionally, as we mention and discuss in the manuscript, we already demonstrated that PDE4 mRNA and protein expression is increased in livers of AH patients when compared to controls/donors (PMID: 31081957). We could provide our qPCR results if needed.

“10. In the material and method section, the details about bioinformatics analysis shall be provided.”

Response: We have expanded the Methods section 2.9. RNA sequencing for better clarification of the approach used in the study and for data analysis as well.

“11. Caption of figures must be self-explanatory. Currently, the caption of figures is very superficial.”

Response: We have expanded the figure legends to clarify the nature of the assays and the type of experiment being conducted.

Round 2

Reviewer 2 Report

All the concerns have been addressed.